# Functional Analysis of *PbbZIP11* Transcription Factor in Response to Cold Stress in Arabidopsis and Pear

**DOI:** 10.3390/plants13010024

**Published:** 2023-12-20

**Authors:** Yuxin Zhang, Lin Wu, Lun Liu, Bing Jia, Zhenfeng Ye, Xiaomei Tang, Wei Heng, Li Liu

**Affiliations:** College of Horticulture, Anhui Agricultural University, 130 Changjiang West Road, Hefei 230036, China; zhangyuxin@stu.ahau.edu.cn (Y.Z.); wulin0205@stu.ahau.edu.cn (L.W.); liulun@ahau.edu.cn (L.L.); jb1977@ahau.edu.cn (B.J.); yezhenfeng@ahau.edu.cn (Z.Y.); tangxiaomei@ahau.edu.cn (X.T.)

**Keywords:** pear, cold stress, *PbbZIP11*, CBF-dependent signaling

## Abstract

Cold stress is a prominent abiotic factor that adversely affects the growth and yield of pears, consequently restricting the cultivation range and resulting in substantial economic losses for the pear industry. Basic region–leucine zipper (bZIP) transcription factors are widely involved in multiple physiological and biochemical activities of plants, particularly in response to cold stress. In this study, the responsiveness of *PbbZIP11* in pear to cold stress was investigated, and its role was explored by using pear callus and *Arabidopsis thaliana*. The findings revealed that overexpression of *PbbZIP11* enhanced the tolerance of pear callus and *Arabidopsis thaliana* to cold stress. The antioxidant enzyme activities of transgenic plants were enhanced and the expression of C-repeat binding transcription factor (CBF) genes was increased as compared to wild-type plants. To better understand the biological function of *PbbZIP11*, mRNAs were isolated from overexpressed and wild-type *Arabidopsis thaliana* after cold stress for whole-genome sequencing. The results showed that the expression of some CBF downstream target genes changed after exposure to cold stress. The results suggested that the *PbbZIP11* gene could participate in cold-stress signaling through the CBF-dependent pathway, which provides a theoretical basis for the *PbbZIP11*-mediated response to cold stress and for the genetic breeding of pear varieties with low-temperature tolerance.

## 1. Introduction

In recent years, there has been a surge in the occurrence of low-temperature disasters, giving rise to a host of cold-related damages such as frost, cold waves, and even untimely springs. These agro-meteorological disasters impose a range of physiological and biochemical effects on pear trees, thereby constraining the advancement of the pear industry [1,2]. Extensive investigations have been undertaken to unravel the intricate mechanisms of cold stress in plants, encompassing both cold damage beyond the 0 °C threshold and freezing damage below it.

Exposure to cold stress engenders a derangement in normal physiological activities within plant cells, adversely impacting the cell membrane system and osmotic substances, as well as the balance of reactive oxygen species (ROS) within the plant microenvironment [3,4]. The intricate process of low-temperature signaling in plants constitutes a remarkably intricate interplay. Under low-temperature conditions, plants sense low-temperature signals through signaling pathways. This prompts the ignition and subsequent regulation of the expression of specific low-temperature-induced genes via transcription factors, which represents a pivotal juncture in plants’ response to cold stress. Genetic engineering has emerged as a demonstrably efficient and time-economical avenue for generating new varieties endowed with superior stress resistance, as compared to conventional breeding practices [5].

Basic region–leucine zipper (bZIP) transcription factors are widely distributed in eukaryotes, characterized by a high conservation with two domains: a basic domain and a leucine zipper domain [6]. Specifically, the N-terminal basic domain (N-X7-R/K-X9) displays significant conservation with approximately 16–20 basic amino acid residues, housing nuclear localization sequences of genes and DNA recognition structural domains that selectively bind to DNA sequences [7]. Conversely, the C-terminal leucine zipper domain, comprising one or more heptapeptide repeats that assemble into one alpha helix, lacks similar levels of conservation. These heptapeptide repeats encompass a leucine or another hydrophobic residue, such as isoleucine, valine, and methionine [8]. The bZIP transcription factors are present in large numbers in plants, with each plant genome boasting distinct members [9]. With comprehensive whole-genome sequencing, the existence of bZIP transcription factors has now been detected in various plants, including *Arabidopsis thaliana*, *Malus domestica*, *Glycine max*, *Zea mays*, *Solanum lycopersicum*, *Oryza sativa*, *Pyrus bretschneideri*, and others. Among them, the model plants have been meticulously scrutinized, 78 AtbZIP transcription factors were identified in *Arabidopsis thaliana* [10], and 92 PbrbZIP genes were identified in the white pear genome [11].

The bZIP transcription factors are involved in a variety of physiological and biochemical processes within the thriving realm of plant life, such as cell stretching, metabolic activity, hormone and sugar signaling, pathogen defense, and the endurance of diverse abiotic stresses. Notably, these transcription factors are particularly significant in responding to the perils of frigid circumstances [12]. The expression of *MdHY5* transgenic apple healing tissues exhibited superior hardiness against icy temperatures, demonstrating that overexpression of *MdHY5* enhances cold tolerance in apple plants [13]. The overexpression of *SlHY5* in tomato orchestrated the regulation of other genes responsive to stresses, thereby affirming the potential of *SlHY5* gene overexpression to bolster the cold resilience of tomato seedlings [14]. Overexpression of *TabZIP14-B* and *TabZIP60* in wheat in Arabidopsis enhances low-temperature tolerance in transgenic Arabidopsis. Arabidopsis plants emboldened with the augmented presence of *CsbZIP6* showcased a greater fortitude against frost, distinctly surpassing the capabilities of wild-type Arabidopsis following periods of low-temperature stress. A wealth of evidence stipulates that the overexpression of transcription factors, as key regulators within intricate signaling networks, offers an avenue of interaction with pertinent cis-acting elements residing in the target gene promoter regions. This intricate interplay exerts a potent countermeasure against the adversities presented by abiotic stressors, and these target genes often act in a synergistic manner [15]. The bZIP transcription factor augments the plant’s capacity to withstand the rigors imposed by chilling temperatures through their regulation of the expression profiles of a multitude of stress-responsive genes, each harboring immense potential as putative targets of the bZIP repertoire. *GmbZIP44*, *GmbZIP62*, and *GmbZIP78* in soybean bind to ABRE elements and activate the expressions of their downstream genes, *ERF5*, *KIN1*, *CORl5A*, and *COR78*, to improve salt and cold tolerance in plants [16]. *OsbZIP52* can specifically bind to the cis-acting element G-box on the promoters of downstream genes and initiate the expression of downstream abiotic genes, such as *OsTPP1*, *OsLEA3*, and *Rab25*. This results in the overexpression of *OsbZIP52*, which reduces the tolerance of rice to cold stress [17].

In this investigation, we meticulously examined the physiological and biochemical functionalities of the *PbbZIP11* gene under cold stress by obtaining a homozygous transgenic pear callus and heterozygous transgenic *Arabidopsis thaliana*. Subsequently, we avidly scrutinized the transcriptome data of wild-type *Arabidopsis thaliana* and transgenic *Arabidopsis thaliana* following chilling conditions with the overarching aim of unearthing their downstream target genes. This primary objective of this study not only sought to unravel the intricate molecular mechanism through which the *PbbZIP11* gene responds to cold stress in pear, but to also furnish a panoply of valuable insights that could pave the way for enhancing the resistance of diverse crops via the modality of transgenic engineering.

## 2. Results

### 2.1. Bioinformatics Analysis of PbbZIP11

The *PbbZIP11* (Pbr026741.1) gene is 474 bp in length. Its open reading frame (ORF) encompasses a total of 474 bases with no non-coding region base sequences. Furthermore, it encodes a total of 157 amino acids with the most abundant being serine (Ser) and aspartic acid (Asn). The relative molecular weight of *PbbZIP11* is 17,900.02, and the isoelectric point is 5.41. The total number of positively charged residues (Arg + Lys) was 21, the total number of negatively charged residues (Asp + Glu) was 18, and the fat index was 67.13.

The NCBI (http://blast.ncbi.nlm.nih.gov/Blast.cgi; accessed on 15 November 2021) database was searched for homologous sequences of *PbbZIP11* in other species and 14 sequences were obtained. The amino acid sequences of the other species were compared with the *PbbZIP11* protein sequence. From the results, it can be seen that the *PbbZIP11* protein has almost the same bZIP conserved structural domain as the other bZIP proteins (Figure 1A). Evolutionary tree analysis using MEGA software showed that the *PbbZIP11* protein is more conserved and has a high similarity with other plants in the Rosaceae family, and its coding amino acid belongs to the same branch as MdbZIP11 in apple, which is the closest relative (Figure 1B).

### 2.2. Expression Analysis of PbbZIP11 in Pear

To determine the expression of the *PbbZIP11* gene under low-temperature conditions, we determined the relative expression of the *PbbZIP11* gene at different durations of low-temperature in *P. betulifolia* seedlings. The expression of the *PbbZIP11* gene increased with the increase in the low-temperature duration and peaked at 12 h, which was 3.2 times the initial value (Figure 2A). This indicates that the *PbbZIP11* gene could play an important role in low-temperature stress in pear plants.

The expression pattern showed that the *PbbZIP11* gene was expressed in the roots, shoots, flowers, leaves, young fruits, fruits flesh, and pericarp of pear plants. Using the expression of the *PbbZIP11* gene in root tissues as a control, it can be seen that there is no significant difference in the expression of the target gene in the buds, young fruits, and mature fruits, and it is significantly expressed in the flowers, leaves, and pericarp. The expression in pear leaves was 7.6 times higher than that in the roots, whereas the expression of the *PbbZIP11* gene reached its highest value in the flowers and pericarp, which were 27.1 and 28.2 times higher than that in the roots, respectively (Figure 2B).

### 2.3. Overexpression of PbbZIP11 Improves Resistance to Cold Stress in Transgenic Arabidopsis thaliana

Since *PbbZIP11* was observed to be responsive to cold stress, we subsequently explored the function of the *PbbZIP11* gene using transgenic material. We successfully constructed an overexpression construct of *PbbZIP11* (Figure 2C). We extracted RNA from wild-type and transgenic Arabidopsis and determined the expression of the *PbbZIP11* gene in different strains by RT-qPCR. The expression of the *PbbZIP11* gene was significantly higher than that of the wild-type Arabidopsis in five of these Arabidopsis strains, which proved that the *PbbZIP11* gene was successfully transfected into Arabidopsis. For ease of expression, subsequent overexpression plants were denoted by OE. OE2, OE3, and OE4 with higher expression levels were selected to continue breeding to the T_3_ generation to obtain pure transgenic Arabidopsis; thus used for the subsequent functional characterization of the gene (Figure 2D). To verify the resistance of transgenic Arabidopsis to cold stress, we first examined the effect of low temperature on seedling root growth. There was no significant difference in root growth between wild-type and transgenic Arabidopsis at room temperature (25 °C). However, when seedlings were subjected to cold stress (4 °C), the roots of transgenic Arabidopsis were significantly longer compared with those of the WT lines. Moreover, it is known that low temperatures induced root growth retardation in both WT and transgenic plants, and that the effect was more pronounced in WT seedlings (Figure 3A,B). After seedlings were subjected to cold stress at −20 °C for 1 h, the average survival rate of transgenic Arabidopsis reached 40.4%, which was approximately 3.1 times higher than that of wild-type Arabidopsis (Figure 3C,D).

Under normal conditions, WT and transgenic Arabidopsis did not differ significantly in morphology. When they were subjected to cold stress, the injury appeared mainly in the leaves of WT plants. After 3 days of recovery at 25 °C, most of the leaves of WT plants wilted (Figure 3E). Overall, transgenic Arabidopsis was more likely to recover to growth after cold stress than WT Arabidopsis. In addition, we performed histochemical staining with diaminobenzidine (DAB) and nitrogen blue tetrazolium (NBT), to show the levels of H_2_O_2_ and O_2_^-^ accumulation, respectively. The results showed that when subjected to cold stress, both transgenic and WT Arabidopsis exhibited deeper staining, with the WT strain exhibiting deeper and more intense DAB and NBT staining compared with transgenic Arabidopsis (Figure 3F). Plants subjected to cold stress exhibited different degrees of damage to the membrane system, and the MDA content of plants can indirectly show the degree of damage to the cell membrane system of plants under cold stress [18]. We then measured the MDA content in plants, and the transgenic Arabidopsis exhibited lower MDA levels compared with the wild type after cold stress (Figure 3G). These data suggest that *PbbZIP11* may attenuate the accumulation of ROS in vivo and inhibit cell membrane damage after plants were subjected to cold stress.

### 2.4. Overexpression of PbbZIP11 Improves the Resistance of Transgenic Pear Calli to Cold Stress

We extracted mature pear callus RNA and selected five transgenic pear calluses for qRT-PCR analysis. We selected OE1, OE2, and OE3, which had higher expression compared with the control, to be cultured in successive generations until we obtained the T_3_ generation of the pear callus for subsequent experimental analysis (Figure 4A). Pear calluses that were as large as possible were selected and spread flat into an antibiotic-free MS solid medium, and put into a room-temperature incubator (25 °C) or low-temperature incubator (4 °C), in the absence of light, for two weeks for observation. The results showed that there was no significant difference between the calluses grown at room temperature, and the growth of all of the calluses was inhibited in the low-temperature incubator. However, the difference in their growth observed by the naked eye was not significant (Figure 4B). Consequently, we measured the fresh weight of the pear calluses. The results showed that the fresh weight of OE1 was not significantly different from that of the WT pear calluses after low-temperature treatment, whereas the fresh weight of OE2 and OE3 was significantly higher than that of the WT pear calluses (Figure 4C).

### 2.5. Overexpression of PbbZIP11 Enhances Antioxidant Enzyme Activity in Plants after Low Cold Stress

The strength of plant resistance to stress is closely related to the ability of its in vivo antioxidant system to scavenge ROS. In order to further validate the cold stress tolerance of the transgenic plants, the in vivo antioxidant enzyme activities of the plants were assayed. The results showed that there was no significant difference in the activities of CAT, SOD, and POD in both WT and transgenic plants under normal conditions. However, after being subjected to cold stress, the activities of all three antioxidant enzymes in the plants increased significantly, and the activities of all three antioxidant enzymes were significantly higher in the transgenic plants compared to the WT plants (Figure 5A–C). It was demonstrated that overexpression of the *PbbZIP11* gene increased the activity of antioxidant enzymes in vivo in *Arabidopsis thaliana* and pear calluses (Figure 5D–F), thereby enhancing plant resistance to cold stress.

### 2.6. Overexpression of PbbZIP11 Enhances the Expression of Low-Temperature-Related Genes in Plants after Cold Stress

When plants are subjected to cold stress during growth, the expression of some relevant genes responding to low-temperature signaling in their bodies will be up-regulated, and the cold resistance of plants will be improved through the synthesis of relevant proteins. The transcriptional cascade response regulated by the CBF gene family is an important response to low-temperature signaling. To understand the potential mechanism of altered cold resistance in *PbbZIP11* transgenic plants, the expressions of CBF genes in WT and transgenic plants after cold stress were examined by qRT-PCR. The results showed that overexpression of *PbbZIP11* could enhance the expressions of CBF genes in transgenic plants after cold stress (Figure 6A–C). In particular, both CBF1 and CBF3 were expressed at significantly higher levels in *PbbZIP11*-overexpressing pear calluses and in Arabidopsis than in the wild type (Figure 6D–F). As expected, overexpression of *PbbZIP11* positively regulates the CBF gene.

### 2.7. Transcriptome Analysis on WT and Transgenic Arabidopsis thaliana after Cold Stress

To obtain the expression profiles of Arabidopsis genes differentially regulated by *PbbZIP11* under cold stress, three biological replicates of each of WT Arabidopsis and transgenic Arabidopsis after cold stress were sent for transcriptome sequencing, totaling six samples. A total of 37.84 Gb clean reads were obtained, and the clean reads of each sample reached more than 5.84 Gb. The clean reads were compared with the Arabidopsis genome, and the percentages of total reads with multiple comparison positions on the reference genome of the total number of clean reads (total mapped) were all above 90% (Appendix A). In addition, the percentages of total reads with a single comparison position on the reference genome of the total number of clean reads (uniquely mapped) ranged from 88% to 94% (reference genome source: http://plants.ensembl.org/Arabidopsis_thaliana/Info/Index; accessed on 8 June 2022).

Based on the quantitative results of the sample expression for intergroup differential gene analysis, a total of 1379 differentially expressed genes were screened between transgenic *Arabidopsis thaliana* and wild-type *Arabidopsis thaliana* with |Log2FC| ≥ 1 and *p*-adjust < 0.05 as the screening thresholds. Among these, 764 genes were up-regulated and 615 were down-regulated. Functional categorization of differentially expressed genes based on GO annotation and official classification shows that most of the DEGs belonging to the ‘biological processes’ category have functions in bioregulation, metabolic processes, response to stimuli, and cellular processes (Figure 7A). In addition, comparison of the differentially expressed genes with the KEGG database showed that the most abundant annotations were in the ‘plant–pathogen interaction’, followed by ‘plant hormone signal transduction’ and the ‘MAPK signaling path’. KEGG pathway analysis indicated that *PbbZIP11* responds to cold stimuli by exerting influence through a series of signaling processes (Figure 7B).

### 2.8. Expression of Related Cold-Responsive Genes in Arabidopsis thaliana

Plants usually respond to cold stress through CBF-dependent pathways. We selected five already-reported CBF downstream target genes (*AtRD29a*, *AtK1N1*, *AtCOR47*, *AtCOR15a*, *AtCOR413*) from the transcriptome data. As is shown in the results, the expression of these genes was not significantly different in both transgenic Arabidopsis and wild-type Arabidopsis under normal conditions. After being subjected to cold stress, the expression of all these genes in transgenic Arabidopsis increased. This was significantly higher than that in wild-type Arabidopsis (Figure 8), and the results were consistent with RNA-seq. Thus, the higher tolerance of *PbbZIP11* transgenic Arabidopsis to low temperature may depend, in part, on the increased expression of these genes.

## 3. Materials and Methods

### 3.1. Plant Material and Growing Conditions

‘Huangguan’ pear, as one of the many characteristic pear varieties in China, with a large planting area and excellent comprehensive traits, is favored by consumers. However, it has been seriously affected by cold damage in recent years. ‘*Pyrus betulaefolia*’ is the rootstock of the ‘Huangguan’ pear. Therefore, we chose the ‘Huangguan’ pear and ‘*Pyrus betulaefolia*’ as the experimental materials. In order to analyze the expression pattern of *PbbZIP11*, 8-week-old ‘*Pyrus betulaefolia*’ tissue culture seedlings with the same amount of growth were treated at a low temperature of 4 °C for 0, 3, 6, 9, and 12 h. Three biological replicates were set up for each treatment, and at each time point, 2–3 leaves of the seedlings were selected for sampling and immediately placed in liquid nitrogen for rapid freezing. In terms of tissue expression, 25-day-old ‘Huangguan’ fruits were used as young pear fruits; 100-day-old ‘Huangguan’ pear fruits were used as mature fruits, and flowers, leaves, and stems were obtained from ‘Huangguan’ pear trees. The above materials were frozen in liquid nitrogen and stored at −80 °C in a refrigerator. The RNAs of these samples were extracted for subsequent experiments.

### 3.2. Sequence Analysis and Gene Cloning of PbbZIP11

The *PbbZIP11* gene (Accession number: LOC103937805) was obtained from NCBI, which encompasses a total of 474 bp in its open reading frame, with no non-coding-region base sequences, encoding a total of 157 amino acids. The conserved structural domains were predicted using SMART (Appendix A), and the 26 bp to 90 bp of *PbbZIP11*, consisting of a leucine zipper and a basic amino acid region, represent the signature structural domain of the bZIP family. Meanwhile, a phylogenetic tree of *PbbZIP11* and bZIP11 proteins of other species was constructed with MEGA5.0 software.

RNA extraction was performed using the RNAprep Pure Plant Plus Kit (DP432- TIANGEN, Beijing, China). Then, using RNA as a template, the first-strand cDNA was synthesized using TransScript^®^ First-Strand cDNA Synthesis SuperMix (AT301-TRAN, Beijing, China). The CDS region of *PbbZIP11* was used as the reference sequence, and primers were designed using Primer 5.0 software (Appendix A) to amplify *PbbZIP11*. Purified DNA was ligated into the cloning vector pEASY^®^-Blunt Simple Cloning Kit (CB111-TRAN, Beijing, China) and sequenced.

### 3.3. PCR Analysis

A qRT-PCR experiment was conducted to identify the transgenic plant materials and examine the expression levels of cold-responsive genes in transgenic plant. The qRT-PCR experiment was conducted using the 2 × SYBR Green qPCR Mix (PC3301-Aidlab, Beijing, China) according to the manufacturer’s instructions. All reactions were run as three technical replicates on a QuantStudio 5 Real-Time PCR System. All of the primers used are shown in Appendix A.

### 3.4. Acquisition of Transgenic Plant Materials

The over-expression vector *PbbZIP11*-pCAMBIA1300 was constructed by inserting the ORF of the *PbbZIP11* gene into the transformed vector pCAMBIA1300. The primers used for vector construction are shown in Appendix A. Transgenic pear calluses and transgenic Arabidopsis were generated through Agrobacterium-mediated genetic transformation [19]. The infested pear calluses were cultured on MS medium supplemented with 0.5 mg/L 6-BA and 1 mg/L 2,4-D at room temperature and protected from light. Screening with kan+ resistance plates was carried out after 45–60 days, following by sampling for identification. *Arabidopsis thaliana* was grown at room temperature by inflorescence infestation followed by seed spotting into MS medium, and then moved into the soil after 15 days for normal culture. The plant material was sampled for identification after it had matured.

### 3.5. Low-Temperature Stress Treatment of Transgenic Plant Materials

In order to treat pear calluses with cold stress, the calluses from 8 d after succeeding generations were placed in a light-protected culture at 4 °C for two weeks for observation and sampling [20]. The fresh weights of pear healing tissues were determined at the same time.

Arabidopsis root observations were carried out by exposing one-week-old seedlings to a low temperature of 4 °C for 7 d [21]. Viability observations were carried out by exposing two-week-old seedlings to a low temperature of −4 °C for 1 h [22], and they were later incubated for 7 d under normal growth conditions. Arabidopsis phenotypic observations after transplantation were carried out by exposing six-week-old Arabidopsis plants to a low temperature of 4 °C for 8 h and then taking samples to preserve them.

All experiments were set up with control and experimental groups, with at least three biological replicates for each treatment.

### 3.6. Measurement of Relevant Physiological Indicators

Reactive oxygen histochemical staining was carried out by taking fresh leaves from the plants, washing them with distilled water, and immersing them in DAB or NBT staining solutions. Then, the test tubes were wrapped in aluminum foil and left at room temperature overnight, whereas chlorophyll was later removed by heating the leaves by immersing them in ethanol. MDA content was measured using a colorimetric method with a spectrophotometer. Antioxidant enzymes such as POD, SOD, and CAT activities were measured using Solepol kits.

### 3.7. Transcriptome Analysis of Transgenic and WT Arabidopsis thaliana

In this experiment, six samples, comprising three biological replicates each of WT *Arabidopsis thaliana* and transgenic *Arabidopsis thaliana* after 8 h of cold stress were sent for transcriptome sequencing, totaling six samples. RNA extraction, library preparation, and deep sequencing were performed at Shanghai Meiji Biomedical Technology Co. (Shanghai, China).

All transcripts obtained by sequencing were aligned with the TAIR10 database for functional annotation. The FPKM value was determined for the expression level of each single gene. Differential gene analysis was performed between transgenic and WT plant groups based on the quantitative results of sample expression, and DEGs were screened using |Log2FC| ≥ 1 and *p*-adjust < 0.05 as the screening thresholds (Appendix A). DEGs were functionally categorized according to GO annotation and the official classification (*p*-adjust < 0.05) (Appendix A). The KEGG (Kyoto Encyclopedia of Genes and Genomes) database, used for annotation, also showed significantly enriched pathways (*p*-adjust < 0.05) (Appendix A).

### 3.8. Data Analysis

Graphs were plotted using GraphPad Prism 6.0 software and analyzed for significant differences using SPSS 18.0 statistical software. The results of RT-qPCR were calculated using the 2^−∆∆CT^ method before analysis.

## 4. Discussion

Throughout the extensive trajectory of evolution, plants have developed an intricate array of mechanisms to confront and adapt to formidable adversities, encompassing the perception of stress signals and modifications in physiological and biochemical responses. Among these mechanisms, the significance of transcription factors, particularly belonging to the bZIP family, are paramount in facilitating plant resilience against the rigors of low-temperature stress. During the initial phases of stress incursion, genes exhibit swift reactions to the external milieu, attaining zenith expression levels mere hours following induction, only to subsequently dwindle [23]. It was shown that in wild-type Arabidopsis, *AtbHLH122* expression reached a maximum at 12 h of cold stress [24]. In this study, the expression of the *PbbZIP11* gene showed a gradual increase after cold stress in pear and peaked at 12 h. These bZIP transcription factors are anticipated to play significant roles in the intricate interplay of plant growth and development. Within Arabidopsis, the *AtbZIP11* gene exerts its influence over the root growth regulator IAA3, thus exerting control over root growth. In this study, the *PbbZIP11* gene manifested its highest degree of expression within the captivating realms of pear blossoms and the pericarp. This exceeded the level witnessed in the roots by prodigious proportions, with a 27.1-fold difference in the blossoms and 28.2-fold difference in the pericarp, to be exact. Notably, transgenic plants, when subjected to low-temperature stress, demonstrated heightened tolerance and adaptability, in stark contrast to their wild-type counterparts. This observation potently hints that *PbbZIP11* could conceivably stand as an influential conductor of cold-stress response.

ROS represent vital byproducts generated amidst the metabolic intricacies of higher plants, intricately entwined with pivotal physiological and biochemical phenomena. Typically, plants produce a certain amount of ROS during their daily growth and metabolism. However, at the same time, antioxidant defense mechanisms are working to scavenge the excess ROS as a way of achieving a dynamic balance between ROS production and scavenging in plants [25]. When plants are confronted with the chilling embrace of cold stress, the activities of various antioxidant enzymes are significantly enhanced, which improves the efficiency of scavenging reactive oxygen species and stabilizes ROS at appropriate levels [26]. These enzymes are involved in the regulation of endogenous and exogenous cellular signaling pathways and play a crucial role in plant defense against abiotic adversity. Consistent with the prior literature, our findings revealed no substantial disparity in the activities of antioxidant enzymes between transgenic Arabidopsis and its wild-type counterpart under normal circumstances. However, transgenic Arabidopsis displayed significantly heightened levels of CAT, SOD, and POD activities compared to WT Arabidopsis under cold stress. This observation suggests that transgenic Arabidopsis possesses a greater capability to scavenge superoxide ions and exhibits enhanced resilience to cold temperatures. The activities of antioxidant enzymes exhibited dynamic fluctuations over the course of time following oxidative stress induction. It was shown that the antioxidant enzyme activities in the healing tissues of avocado and snow pear exhibited different changes after NaCl treatment in vivo. The in vivo SOD and POD activities in the healing tissues of *P. betulifolia* showed a trend of increasing and then decreasing, while CAT activity showed a trend of decreasing and then increasing. The in vivo SOD and POD activities in the healing tissues of *P. betulifolia* showed a trend of decreasing and then increasing, reaching the peak at 12 h. In this study, the POD, SOD, and CAT activities in the transgenic pear’s healing wounds failed to show significant divergence from those in the wild type under normal circumstances. However, following 6 h of cold stress, the antioxidant enzyme activities within the healing tissues underwent variable alterations, with the POD, SOD, and CAT activities in the transgenic pear’s healing wounds significantly surpassing those in the control group. The variations witnessed in the antioxidant enzyme activities following low-temperature treatment in Arabidopsis and pear guaiac signify that the overexpression of the *PbbZIP11* gene can facilitate their adaptation to cold environments by reinforcing the antioxidant defense system’s capacity in these organisms in vivo.

The ability of a plant to withstand cold or frost is contingent upon the extent to which stress genes are expressed during the process of developing resistance and adapting to cold-induced stress. These genes intricately weave a molecular regulatory network that fortifies the plant against the perils imposed by adversity. Functionally speaking, these stress-responsive genes can be classified into two distinct categories: functional genes, which directly combat the deleterious consequences of adversity, and regulatory genes, which orchestrate the intricate dance of gene expression in response to stress [27]. Transcription factors belong to regulatory genes that can regulate the expression of a large number of downstream genes, thereby altering plant stress tolerance. An abundance of research has unveiled the pivotal role played by three members of the AP2/ERF family of transcription factors in the cold response of *Arabidopsis thaliana*, namely CBF1, CBF2, and CBF3, collectively referred to as DREB1/CBFs. These genes have been gradually verified to be important for cold training in other species [28]. When confronted with abiotic stresses, plants deftly bolster their resistance by orchestrating the interplay between transcription factors and cis-elements encoded within the promoter regions of genes associated with the specific stress at hand. This intricate symbiosis, facilitated by internal signaling pathways, engenders the precise regulation of an array of stress-related genes, amplifying the plant’s fortitude in the face of adversity. MbMYB4 in apple plants promotes the expression of *AtCBF1* and *AtCBF3* in transgenic *Arabidopsis thaliana* after cold stress [29]. The bZIP structural domains of bZIP-type transcription factors preferentially bind to motifs such as the A-box (TACGTA), C-box (GACATC), and G-box (CACGTG) elements of downstream target genes. Remarkably, it has been demonstrated that the bZIP transcription factor can avidly bind to the CBF promoter region [30]. While prior empirical evidence had highlighted an augmented cold resistance in transgenic pear guaiac and transgenic Arabidopsis via *PbbZIP11* overexpression, the precise mechanism remained elusive. To elucidate the underlying molecular mechanism behind the involvement of *PbbZIP11* in cold resistance, an exploration of its potential role was undertaken through qRT-PCR analysis. Under normal conditions, there was no significant difference in the content of *PbCBF1-3* in both transgenic and wild-type pear healing wounds. However, upon exposure to cold stress, the expression of *PbCBF1-3* in transgenic pear healing wounds was significantly higher than that in wild-type pear healing wounds. The expression of *AtCBF1*, *AtCBF2*, and *AtCBF3* in Arabidopsis was determined with the same results as in pear healing. It was demonstrated that overexpression of the *PbbZIP11* gene could promote the expression of CBF genes in pear healing and Arabidopsis.

It has been shown that CBF genes induce the expression of downstream cold-responsive (COR) genes such as CORs, RDs, and LTIs by binding to CRT/DRE cis-acting elements to enhance cold resistance in plants [31]. This mode of regulation is known as the CBF-dependent pathway [32]. In Arabidopsis, a number of COR genes are involved in cold stress through the CBF/DREB1 pathway. *COR15a* is a cold-inducible gene identified from *Arabidopsis thaliana*. Key genes in the CBF-dependent pathway, CBF1 and CBF3, can bind to the CRT/DRE cis-acting element of *COR15a*, and overexpression of the gene in *Arabidopsis thaliana* leads to enhanced tolerance to freezing stress in transgenic plants [33]. *COR413* and *COR47* belong to a group of poorly characterized plant-specific cold-regulated genes, originally identified as part of the plant transcriptional activation machinery during cold acclimation. The CBF gene in potato is rapidly activated at low temperatures. In turn, this induces the expression of a set of downstream cold-responsive genes, *COR47* and *COR413*, thereby enhancing the freezing tolerance of potato plants [34]. *RD29a*, a key gene in the Arabidopsis ABA signaling pathway, contains two cis-acting elements, one of which is DRE (dehydration response element), which responds to drought, high salinity, and low-temperature stress [35]. The promoter of *KIN1* contains conserved CRT/DRE motifs, which are cold-responsive target genes downstream of CBF, and can be induced by CBF genes, thereby enhancing plant cold tolerance [36]. Although CBF and its target genes are critical for freezing tolerance, only a small fraction of genes are regulated by CBF. In this experiment, the expression of *RD29a*, *KIN1*, *COR47*, *COR15a*, and *COR413* in transgenic Arabidopsis and wild-type Arabidopsis was significantly increased compared with that in wild-type Arabidopsis following cold stress in the transgenic Arabidopsis. It was shown that *PbbZIP11* could regulate the expression of its downstream cold-responsive genes (*RD29a*, *KIN1*, *COR47*, *COR15a*, *COR413*) by affecting the expression of CBF genes. In summary, *PbbZIP11* can participate in low-temperature signaling through the CBF pathway, and the specific regulatory mechanisms need to be verified by further experiments.

## 5. Conclusions

In order to analyze the function of *PbbZIP11*, the transgenic pear healing tissues and Arabidopsis thaliana were subjected to cold stress after obtaining the transgenic pear calluses. The related physiological and biochemical indexes were detected for analysis. The results showed that the overexpression of *PbbZIP11* enhanced the tolerance of the transgenic lines after cold stress. Genome-wide sequencing analysis of mRNA isolated from cold-stressed *Arabidopsis thaliana* showed that the expression of some CBF downstream target genes (*RD29a*, *KIN1*, *COR47*, *COR15a*, *COR413*) related to cold response was significantly elevated by cold stress. We therefore concluded that the *PbbZIP11* gene could be involved in cold-stress signaling through the CBF pathway. This paper provides a reference for the *PbbZIP11*-mediated response mechanism to cold stress and a theoretical basis for the development of low-temperature-tolerant pear varieties.

## Figures and Tables

**Figure 1 plants-13-00024-f001:**
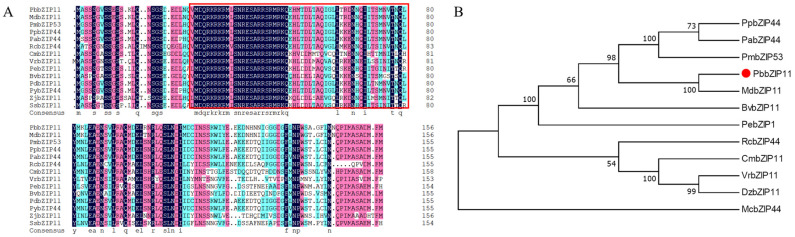
Bioinformatics analysis of *PbbZIP11*. (**A**) Sequence comparison analysis of *PbbZIP11* with other species (*PpbZIP44*: *Prunus persica*, XP_007202732.2; *PdbZIP11: Prunus dulcis*, KAI5319637.1; *PabZIP44: Prunus avium*, XP_021833579.1; *PybZIP44*: *Prunus yedoensis*, PQQ04057.1; *PmbZIP53*: *Prunus mume*, XP_016651997.1; *MdbZIP11*: *Malus domestica*, XP_028946743.1; *BvbZIP11: Bauhinia variegata*, KAI4317459.1; *ZjbZIP11*: *Ziziphus jujuba*, XP_015889898.1; *RcbZIP44*: *Rosa chinensis*, XP_024181466.1; *CmbZIP11*: *Castanea mollissima*, KAF3942600.1; *VrbZIP11*: *Vitis riparia*, XP_034677450.1; *PebZIP11*: *Populus euphratica*, XP_011037478.1; *SsbZIP11*: *Salix suchowensis*, KAG5247795.1). Red boxes indicate bZIP structural domains. (**B**) Phylogenetic tree analysis of *PbbZIP11* with other species.

**Figure 2 plants-13-00024-f002:**
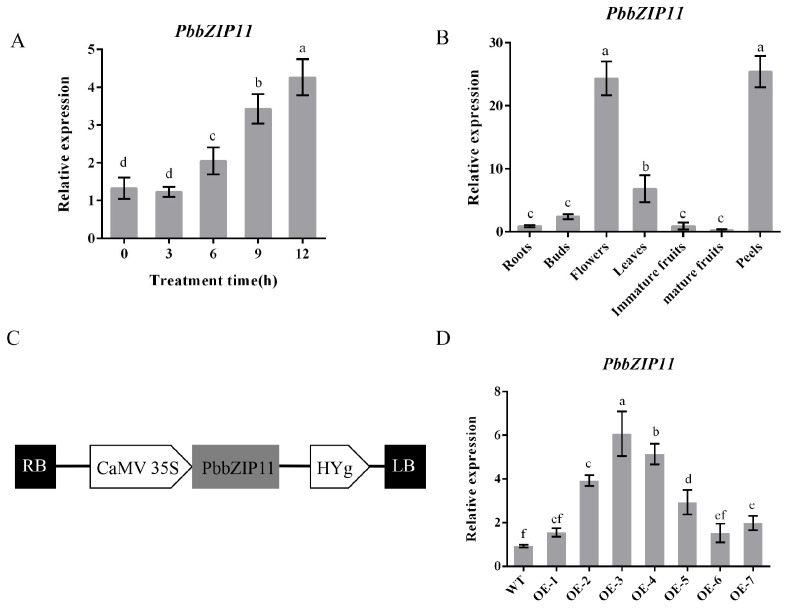
Determination of the expression of *PbbZIP11*. (**A**) The relative expression of *PbbZIP11* at different durations of low-temperature exposure of pear. (**B**) The relative expression of *PbbZIP11* gene in different tissues of pear. (**C**) Schematic diagram of the 35S promoter: *PbbZIP11* construct. (**D**) *PbbZIP11* expression in *Arabidopsis thaliana*. Lowercase letters indicate significant differences between treatments at the *p* < 0.05 level.

**Figure 3 plants-13-00024-f003:**
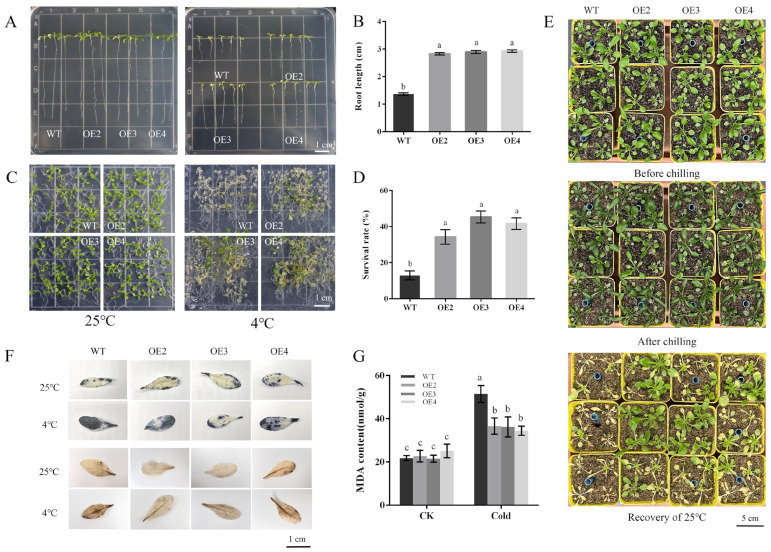
*Arabidopsis thaliana* phenotypic observation. (**A**) Effects of cold stress on *Arabidopsis thaliana* roots. (**B**) Arabidopsis root length measurements. (**C**) Effects of cold stress on *Arabidopsis thaliana* survival rate. (**D**) Survival rate in *Arabidopsis thaliana*. (**E**) Effects of cold stress on *Arabidopsis thaliana* phenotype. (**F**) Effects of cold stress on *Arabidopsis thaliana* reactive oxygen species. (**G**) Contents of MDA under cold stress. Lowercase letters indicate significant differences between treatments at the *p* < 0.05 level.

**Figure 4 plants-13-00024-f004:**
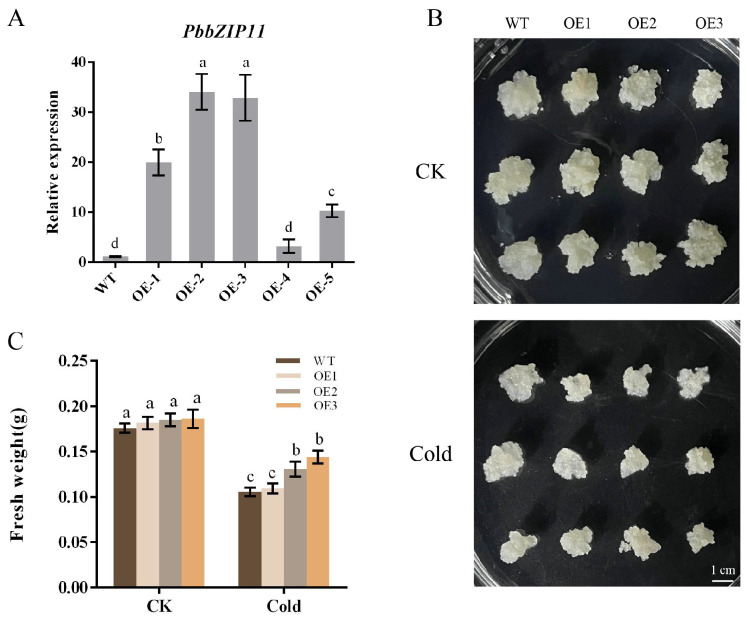
Effect of low temperature on the growth of pear calluses. (**A**) *PbbZIP11* expression in pear calluses. (**B**) Observations of pear callus phenotypes. (**C**) Determination of callus growth in pear. Lowercase letters indicated significant differences between treatments at the *p* < 0.05 level.

**Figure 5 plants-13-00024-f005:**
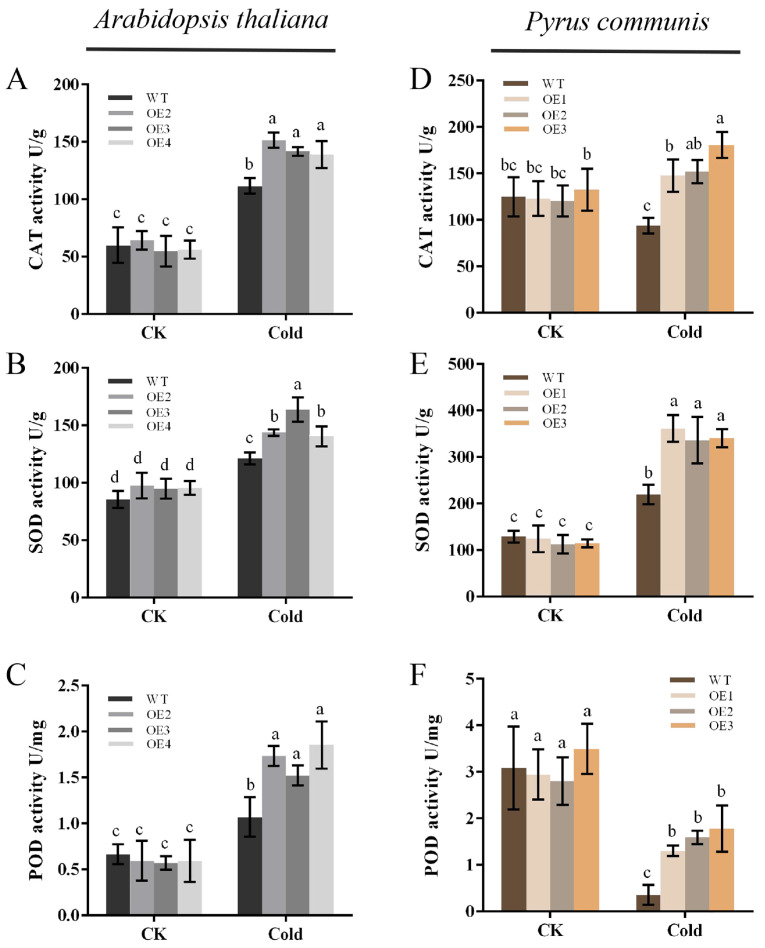
Overexpression of *PbbZIP11* increases antioxidant enzyme activities in transgenic plants after cold stress. (**A**–**C**) CAT, SOD, and POD activities in *Arabidopsis thaliana*. (**D**–**F**) CAT, SOD, and POD activities in pear calluses. Lowercase letters indicate significant differences between treatments at the *p* < 0.05 level.

**Figure 6 plants-13-00024-f006:**
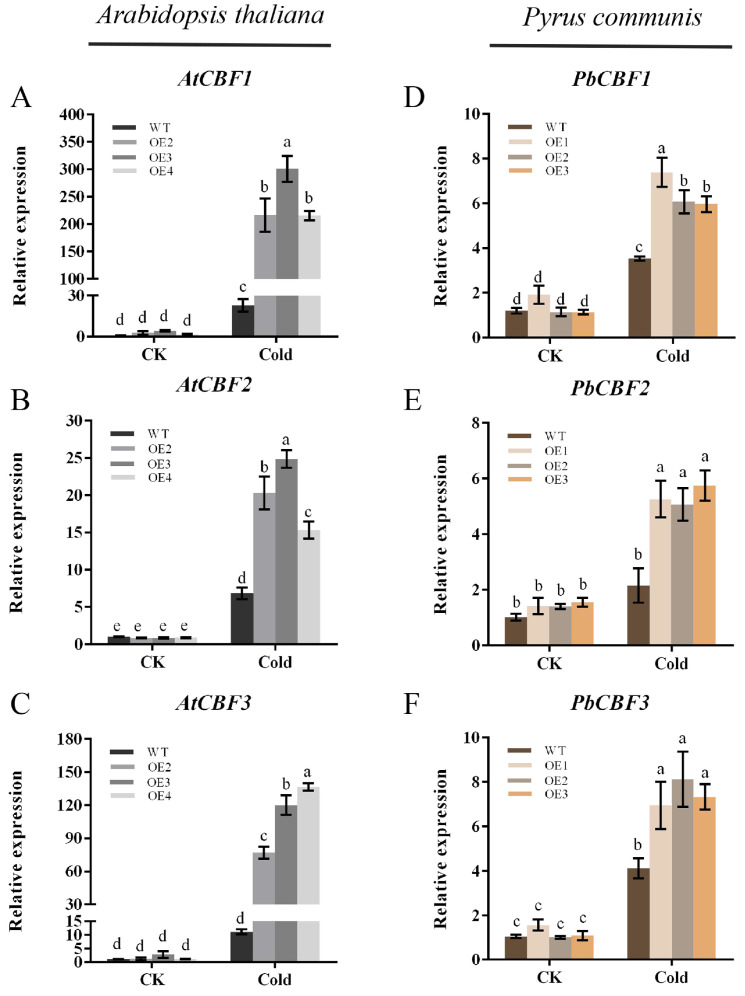
Overexpression of *PbbZIP11* increases low-temperature-related genes in transgenic plants after cold stress. (**A**–**C**) Expression of *AtCBF1*, *AtCBF2,* and *AtCBF3* in *Arabidopsis thaliana*. (**D**–**F**) Expression of *PbCBF1*, *PbCBF2*, and *PbCBF3* in pear. Lowercase letters indicate significant differences between treatments at the *p* < 0.05 level.

**Figure 7 plants-13-00024-f007:**
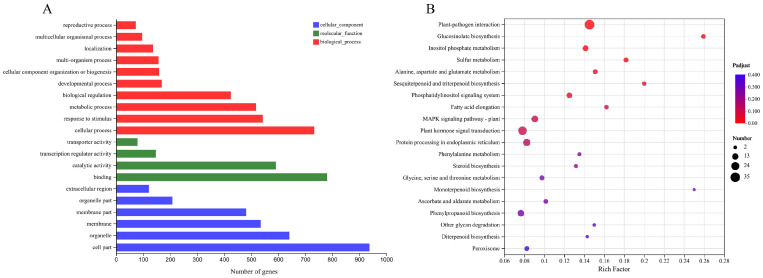
Transcriptome analysis on WT and transgenic *Arabidopsis thaliana* after cold stress. (**A**) GO annotation analysis of differentially expressed genes. (**B**) KEGG analysis of differentially expressed genes.

**Figure 8 plants-13-00024-f008:**
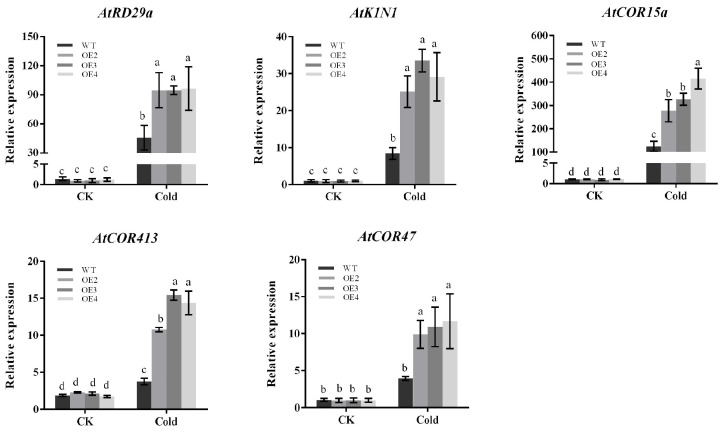
Expression of differential genes in *Arabidopsis thaliana*. Expression analysis of CBF downstream target genes (*AtRD29a*, *AtK1N1*, *AtCOR47*, *AtCOR15a*, *AtCOR413*). Lowercase letters indicate significant differences between treatments at the *p* < 0.05 level.

## Data Availability

The data that support the results of this study are available from the corresponding author upon reasonable request.

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
