# Peer review of "Functional Analysis of PbbZIP11 Transcription Factor in Response to Cold Stress in Arabidopsis and Pear"

_plants, 2023, doi:10.3390/plants13010024_

Round 1

Reviewer 1 Report

Comments and Suggestions for Authors

Comments and Suggestions for Authors

Dear Author,

It is my pleasure to review the manuscript entitled “Functional analysis of PbbZIP11 transcription factor in response to pear cold stress” a research article submitted to MDPI Journal, plants. Authors of this manuscript characterized cold stress responsive transcription factors PbbZIP11 in Pear and Arabidopsis using a series of experiments. Overall, the experiments, they performed, are well and the results are convincing. Thus, the presented results take up an important topic consistent with the profile of the Journal.

However, I have some suggestions, which might improve the manuscript to make important to the wider readers.

·         Improvement in English is necessary for clear understanding

·         I have corrected some point in the main text, please follow them as well

Suggestion for title: Functional analysis of PbbZIP11 transcription factor in response to cold stress in Arabidopsis and Pear

·         The term Arabidopsis thaliana came a number of times. Need to reduce or shortened

·         After kit names please use company name

·         Transgenic selection procedures not mentioned. It is good for followers

·         Many places, throughout the text, space should be inserted

·         For seedling materials, make uniform at the M&M and results sections

·         No description of Fig. 1E before fig. Need to rearrange

·         Fig. 1 title should be more than just Bioinformatic analysis. This figs’ content are not legible. You may divide it in two figs.

·         Selection of transgenic callus is not mentioned

·         Fig. 6. need high resolution

·         L347: suggestion: We selected five already reported CBF downstream target genes (AtRD29a, AtK1N1, 347 AtCOR47, AtCOR15a, AtCOR413) from the transcriptome data

·         Discussion part can be reduced by removing redundant wording and less important descriptions.

·         Conclusion should be added as separate

·         Follow journal style for references

·         Please justify why bbZIP11 genes’ analysis is necessary?

Comments on the Quality of English Language

Minor editing of English language required

Reviewer 2 Report

Comments and Suggestions for Authors

Functional analysis of PbbZIP11 transcription factor in response to pear cold stress

General comment: 

The manuscript is well written and presented. The introduction is focusing on the main topic and on the aims of this study, but the novelty of the study should be underlined. The references are quite recent, include at most works of the last 10 years; However, I have some comments for the improving.

Line 56: “Pyrus breschneideri”: the correct name is: Pyrus bretschneideri

Line 225: it is not clear what are OE2, OE3, as they were not previously mentioned. Abbreviations must be specified in the caption in the right order.

Line 471: “in this experiment” is the correct form.

Supplementary materials are missing header. Moreover, I suggest changing the order of figures and tables.

 Minor considerations:

Affiliations are not reported correctly.

Figure 1 (panels A, C and D), figure 2 and figure 6 are not clearly visible. Please, replace them with high resolved figures. 

Reviewer 3 Report

Comments and Suggestions for Authors

1)    Sequence comparison and phylogenetic tree analyses about Arabidopsis thaliana have not been shown.

2)    POD activity was decreased in transgenic plants of Pyrus communis as compared to wild type. Explain

3)    In fig-3.8, include more genes not regulated by CBF for expression analysis in Arabidopsis thaliana.

4)    Transcriptome analysis data of wild-type and transgenic Arabidopsis thaliana after cold stress treatment needs to be provided in supplementary data.

5)    Phenotypic data of the wild type and overexpression lines include root length and leaf (fig-2). At the same time, the ZIP transcription factor is significantly expressed in flowers relative to roots and leaves (fig-1d), so data regarding its effect on the timing of inflorescence following cold treatment can be provided.

6)    Functional analysis of PbbZIP11 transcription factor in response to pear cold stress.

7)    The abstract should mention the full form of PbbZIP11 and the CBS gene at the 1st instance and then later abbreviations can be used. time use place.

8)    In the material and method, the RNA isolation kit and cDNA synthesis kit catalog with company name.

9)    The material and method, mention the RT-PCR machine company name with the catalog number.

10) In material and method (Plant material and growth condition), why are you choosing Pyrus betulaefolia and Huangguan' pear fruit should be explained.

11) In the result section, a good-quality image of Fig no six should be used.

12) Recheck all the references.

Round 2

Reviewer 3 Report

Comments and Suggestions for Authors

All the queries are answered, and the manuscript is modified accordingly. May be accepted in the present form.

Author Response

Dear Reviewer:
Thank you very much for your kind recognition of our manuscript (plant-2716787) and your valuable comments. Based on the reviewers' suggestions, we made some changes to the introduction and conclusion sections of the manuscript and checked the reference section carefully. We are now sending the revised article. All changes made to the text are shown in red so that they can be easily identified. Please see the full text for details. Thank you again for your constructive suggestions, which are valuable in improving the quality of our manuscript.
Yours sincerely, 
Yuxin Zhang